# Cryoneurolysis: An Emerging Personalized Treatment Strategy for Significant Pelvic Pain

**DOI:** 10.3390/jpm15120587

**Published:** 2025-12-02

**Authors:** Shamar Young, Artyom Abramyan, Ilaria Vittoria De Martini, Jack Hannallah, Gregory Woodhead, Lucas Struycken, Daniel Goldberg

**Affiliations:** Division of Interventional Radiology, Department of Medical Imaging, University of Arizona, Tucson, AZ 85724, USAilariavittoria@arizona.edu (I.V.D.M.); jackh@arizona.edu (J.H.); dgoldberg@arizona.edu (D.G.)

**Keywords:** cryoneurolysis, pudendal nerve, impar ganglion, cancer pain, personalized pain relief

## Abstract

Significant pelvic pain is a fairly common malady in the United States. At the same time, the opioid epidemic has changed how pain is thought about and treated, resulting in a clear call for alternative treatment strategies. One of the promising techniques that has emerged over the last several years is cryoneurolysis. Cryoneurolysis allows for personalization of treatment through targeting of specific peripheral nerves, which correspond to a patient’s pain. In the setting of pelvic pain, several viable targets, namely the pudendal nerve and impar ganglion, have been described. This review delineates the mechanism of action in cryoneurolysis, reviews the pertinent literature and describes patient workup and technique. Finally, future directions are discussed.

## 1. Introduction

The management of pain, and, in particular, chronic pain, has come under significant scrutiny over the last several years [1]. With the onset of the opioid epidemic, a clear call for alternative pain control options has been made [1,2,3,4,5,6,7]. However, pain is a complex and large umbrella term with the cause, location, and type varying greatly. Therefore, solutions also need to be personalized to the pain of a particular patient, and often classifications can be useful when looking at the emerging treatment landscape. While there are numerous ways to classify pain, one useful system is by location and cause. This review will focus on pelvic pain, and we will categorize pain causes into the large, but useful, categories of malignant and non-malignant.

Chronic pelvic pain is a relatively prevalent health care burden, with one study demonstrating a prevalence of 14.7% in a general population of women aged 18–50 [8]. At the same time, another article found that pelvic pain syndromes affect 5.7–26.6% of the population [9]. When turning to cancer-related pelvic pain, studies have found that up to 41.5% of patients have pelvic pain following surgery for colorectal cancer [10]. It is therefore not surprising that in 1996 the estimated direct medical costs for outpatient visits for chronic pelvic pain in women in the United States (US) aged 18–50 were 881.5 million per year [8]. Importantly, but not surprisingly, the quality of life in patients with chronic pelvic pain has been shown to be reduced in both malignant and non-malignant patients [11,12]. Quality of life is affected in a number of ways, with pelvic pain being shown to be associated with anxiety, depression, chronic fatigue, and sleep disorders [13]. This data certainly makes a compelling case for the need for good treatment solutions in this large patient population.

Traditional treatment methods for non-cancerous pelvic pain have included medications [14,15,16], lifestyle modifications [17], peripheral nerve blocks [18,19], pulsed radiofrequency ablation (RFA) [20,21,22,23], peripheral nerve stimulation [24,25,26,27] and surgery [28]. However, all of these approaches have had drawbacks, maintaining the need for further treatment options and innovation. Similar approaches have been taken for cancer-related pelvic pain with comparable results, leading to a well-recognized need for further treatment options in this area as well [29,30]. These factors have combined to form a time of interesting investigation and progress in terms of emerging treatment options and algorithms for chronic pain, in general, and in pelvic pain specifically.

One of the emerging techniques for the treatment of malignant and non-malignant pain is cryoneurolysis. This review evaluates cryoneurolysis and specifically its use in the setting of pelvic pain. The article will review the mechanism of action in cryoneurolysis as a treatment modality, the technique of cryoneurolysis in pelvic targets, and report the available data for this treatment.

## 2. Cryoneurolysis Mechanism of Action

The ability of cold to induce relative numbness has been recognized for centuries, with Hippocrates reportedly using ice and snow to treat painful wounds as early as 460 BC [31,32]. In more recent times, Napoleon’s general surgeon noted that soldiers who were cold had less pain during amputation [31]. However, it was not until the early 20th century that medical devices that could induce freezing temperatures and thus facilitate cryosurgery or cryoablation started to be developed and became available for the treatment of a number of different diseases [33,34,35,36]. The initial uses of these devices did not include peripheral nerve ablation for pain control [33,34,35,36]. Finally, in the 21st century, early reports of percutaneous cryoablation of peripheral nerves for the treatment of pain started to be published [37,38,39,40,41]. Given the relatively novel nature of this approach and technique, it is important for providers to understand the underlying mechanisms of action.

Over time, we have gained a better understanding of peripheral nerve injuries and techniques for their restoration [42,43,44]. Importantly this work has helped us understand the different types or degrees of nerve injury and what this may mean for neuropathic recovery [45,46,47,48]. Sunderland is a frequently utilized nerve injury classification system, and at least a limited understanding of this system is useful when understanding the injury induced by cryoneurolysis and how that differs from other treatment modalities [36,45,46,47,48,49,50,51,52,53]. In brief type I injuries (neurapraxia), results is conduction loss but no structural injury. Type II injuries (axonotmesis) consist of axonal injury. The addition of endoneurium, perineurium and epineurium to the axonal injury takes the injury to type III, type IV and type V, respectively. Of note, cryoneurolysis leads to a Sunderland type II injury, which is fundamentally different from chemical or radiofrequency nerve ablation, which induces a type V injury [54]. A type II injury is a reversible injury, meaning that the nerve will recover function over time [54]. In clinical practice, this means that patients are able to have longer relief of pain than blocks, which typically only last hours to days, but do not have permanent loss of nerve function. The order and rate of nerve regeneration are also predictable [46,47,48]. The neuron will initially undergo Wallerian degeneration, followed by neuron regrowth, typically at a rate of 1–2 mm per day, and finally end organ reinnervation [46,47,48]. What this means in clinical practice is that typically, patients will regain nerve function 3–6 months after undergoing cryoneurolysis.

The mechanism of action in cryoneurolysis may hold some benefits over other similar techniques, such as blocks, chemical neurolysis or RFA [4]. One of those is that the ablation zone can cross anatomic planes, perhaps reducing the burden of probe placement for providers as compared to blocks and chemical ablation. One study that evaluated nerve blocks in patients with confirmed pudendal neuralgia found that it was only effective for approximately 80% of patients [16]. The authors felt that this was largely secondary to technical factors, one of which is that liquid will not typically cross facial planes. Therefore, if the needle tip is close to the nerve but in a different facial plane, blocks and chemical ablations will likely not be effective; however, the same limitation is not present in cryoneurolysis. Cryoablation also offers significant control during the ablation process, as the ablation zone can be visualized through the ice ball formation. The visualization of the ice ball and thus the ablation zone may help to avoid unintended damage to adjacent structures as compared to chemical ablation and RFA. Finally, when compared to RFA, cryoablation may lead to less intra-procedural pain, and there is no concern that the ablation will interfere with pacemakers [4]. However, it is important to remember that while cryoneurolysis holds theoretical benefits over RFA and chemical ablations, it is also an emerging technique with less available data to confirm these advantages, as discussed below.

## 3. Anatomic Targets in the Pelvis

When considering targets for nerve intervention in pelvic pain, there are generally considered to be two: the pudendal nerves and the impar ganglion [29,30,55,56]. Understanding the anatomic course of these nerves and where they can be reliably targeted is an important consideration when discussing the technique of cryoneurolysis. The authors will therefore discuss the anatomy of the pudendal nerves and the impar ganglion.

The pudendal nerves arise from S2–S4 nerve roots and provide sensory as well as motor and autonomic innervation. The sensory innervation is provided to the regions of the anus, perineal and genital regions. The nerve originates proximal to the ischial spine medio-caudal to the sciatic nerve [57,58]. It then passes through the greater sciatic foramen, runs between the sacrospinous and sacrotuberous ligament and twists around the ischial spine posteriorly; finally, it passes through the lesser sciatic foramen and travels through the pudendal (Alcock’s) canal to reach the perineum [57]. When performing cryoneurolysis, the nerve is typically targeted using computed tomography (CT) as it passes through the pudendal canal (Figure 1) [4,37,38]. However, utilizing ultrasound (US) via an ischiorectal approach has also been described in the literature [59].

The impar ganglion transmits sympathetic signals from the perineum, distal rectum, distal vagina, distal urethra and anus and thus makes a good target for pelvic pain therapy [60,61]. The impar ganglion (also referred to as the ganglion impar) is located between the sacrococcygeal joint and coccygeal tip [62,63]. Impar ganglion blocks are often performed under fluoroscopy, through the sacrococcygeal joint, cryoneurolysis is typically performed under CT guidance with a lateral approach (Figure 2) [4,55,62].

## 4. Patient Workup and Technique

While in theory it may seem easy to determine whether a pain source is likely to be within the distribution of the pudendal or impar ganglion distributions, in clinical practice, this can be quite difficult. It is, therefore, vital to spend significant time with the patients and ask focused questions to help determine likely nerve targets. However, even in experienced hands, it may be useful to explain to the patient that if treatment of one nerve site does not lead to the desired results, then treatment of the other target may be worthwhile. When differentiating which nerve target may provide the desired results, the authors focus on the following. Location, anterior or “saddle” area pain is typically best treated with pudendal neurolysis, while posterior pain is typically best treated with the impar ganglion. In patients who have pain exacerbated by sitting, often declining to sit down during a clinic visit, the authors will start with the impar ganglion, while those who have a fear of defecation secondary to pain, the authors will start with the pudendal treatment.

Another key to successful treatment is setting patient expectations. Utilizing a scale to help define the patient’s pain is valuable; the authors prefer the visual analog scale (VAS) [64,65]. Discussing with patients that, particularly in the setting of very high pain scores, a reduction of 50% is a success can help them conceptualize likely treatment outcomes. Furthermore, it may take time for the patients to achieve the full benefit from the procedure. Discussing the concept of pain unveiling can also help patients understand the expected clinical course. Pain unveiling describes the phenomenon of secondary pains becoming more prominent when the primary pain generator is reduced/eliminated. Pain unveiling is particularly common in the setting of cancer pain with multiple metastases, in the author’s experience. If patients are not warned about these things prior to the procedure, then even procedures that the physician may consider to be successful may not be viewed as such by the patient. It is also important to ask pain assessment questions in a targeted manner, that is, the level of pain the patient is feeling in his/her pelvis, and obtain a global pain score to assess efficacy.

The final and critical portion of the pre-treatment consultation, which deserves a bit of discussion, is side effects and adverse events. Adverse events are very rare, but bleeding, infection and damage to adjacent structures are known risks. It is worth noting that, to date, no adverse events have been reported following pudendal cryoablation [37,38,59] or RFA [21,22,23]. It is also important for patients to understand what side effects the procedure may bring. In the setting of pudendal nerve treatments, the authors always inform the patient that they may experience genital numbness and that this may, in turn, lead to sexual dysfunction of sorts. The general idea of numbness of portions of their body is an important concept to relay to patients, as it can be disconcerting to some.

### 4.1. Technique

One of the biggest questions when new procedures emerge is the technique. An important consideration in this setting is whether or not to perform a diagnostic and therapeutic block prior to the cryoneurolysis itself. The authors’ institutional algorithm (Figure 3) classifies patients into inpatients, who cannot be discharged secondary to poor pain control, and outpatients. In inpatients who have a clear cause of pain, such as pelvic malignancy, and also pain in a distribution that corresponds well to a particular nerve target, the authors will tend to recommend proceeding straight to cryoneurolysis to help facilitate discharge. However, if patients do not meet these criteria, the authors will block first, primarily as a diagnostic tool. In outpatients, the authors are more likely to perform a block prior to cryoneurolysis. However, in patients with a clear cause, such as malignancy, who have significant pain leading the authors to feel they are at risk for admission due to poor pain control, they will sometimes recommend that the patient proceed directly to cryoneurolysis. In both cases, we counsel the patients on the positives and negatives of each approach and allow them to choose which option they feel is best. If patients proceed with block prior to cryoneurolysis, then they will be followed up the next day to determine the degree of pain relief. In general, the authors look for a reduction of at least 3 points on the VAS pain scale if they are to recommend cryoneurolysis; however, a thorough review of the pain relief achieved is undertaken with each patient, focusing on the targeted area (also known as pelvic pain).

The authors will perform both impar ganglion and pudendal nerve cryoneurolysis with CT guidance. While pudendal nerve blocks are also performed with CT, ganglion impar blocks are frequently performed with fluoroscopy. For the block, the authors utilize 1–2 mL of lidocaine 1%, 4–7 mL of bupivacaine 0.25%, and 40 mg of Kenalog. In cases where a block is performed, the authors will often have prescheduled the cryoneurolysis to avoid a significant delay between the two procedures. The authors target the pudendal nerve in the pudendal canal and typically try to have the probe oriented to travel along the canal for as much of the placement as possible. While occasionally the authors will treat a single side, they typically will ablate both nerves utilizing two probes (Figure 1). In the case of the impar ganglion, the authors will target it utilizing a tangential approach with a single probe (Figure 2). Pneumo- or hydro dissection may be required to displace the adjacent colon, which the authors often do despite recent evidence suggesting this may not be needed [66]. Freezing protocols have differed significantly, with reported protocols including an 8 min freeze followed by a 5 min passive thaw [38], 8 min freeze followed by a 4 min thaw [37], and a 108 s freeze followed by a thaw [59]. What all reported protocols have in common is that they perform two freeze–thaw cycles and utilize passive thaws. The theoretical reason to avoid warming up the probe with active heat is that heat can lead to agitation of the nerve. All patients are treated with conscious and local sedation at our center.

Patients’ follow-up starts at 1–3 days post-procedure to determine pain relief and then typically at 1, 3 and 6 months. However, patient location (how far away they live), prognosis (many cancer patients may have a life expectancy of <6 months) and response to treatment all play a role in follow-up time tables. When assessing pain improvement, it is important to record overall pain and pain specific to the area of treatment. As discussed above, “unveiling” is often seen, and it is important to determine if the pain they report is related or unrelated to the pain that you were targeting.

Type I nerve injuries have also been reported in cryoneurolysis [4,5,6,67]. Type I injuries lead to increased pain and typically present 2–3 days after the procedure, often with the patient having had an excellent response prior to presenting with increased pain. When this occurs, physicians have two viable approaches. First, they can bring the patient back for a repeat ablation as soon as possible. Second, they can prescribe a short course of steroids, gabapentin and analgesics to help the patient cope with the pain. Type I injuries occur because the targeting of the nerve was suboptimal, and the nerve has reached temperatures that damage it but do not achieve axonotmesis.

### 4.2. Available Literature

While garnering interest, the available literature for cryoneurolysis of pudendal nerves or the impar ganglion is relatively limited (Table 1). In the first report of CT-guided pudendal cryoneurolysis, Prologo et al. described a single-center experience with 11 patients. The average pain prior to treatment on the 10-point VAS was 7.6, which reduced to 2.6, 3.5 and 3.1 at 24 h, 45 days and 6 months post-treatment, respectively (*p* < 0.005 for all) [38]. No post-procedural adverse events (AEs) were reported. Another retrospective single center study of 10 patients with pelvic pain secondary to malignancy demonstrated a mean difference in VAS pain score of 5.2 (*p* = 0.003) [37]. This study also failed to show any significant AEs following treatment. Finally, a case series of two patients treated with US-guided pudendal cryoneurolysis demonstrated a significant reduction in pain for the two patients, without AEs [59].

While significant data exists regarding the treatment of the impar ganglion with RFA or chemical ablation [68,69], to the author’s knowledge, no published data is available on cryoneurolysis of this target. It has been discussed in several review articles [4,5,6]; however, data would be of great value in this area. It is important to note that significant data does exist on both chemical ablation and RFA of both the impar ganglion [68,69] and pudendal [15,16,17,21,22,23,24,25] nerves, and while there are theoretical benefits to cryoablation over these methods, further data on cryoneurolysis is urgently needed.

### 4.3. Future Directions

While cryoneurolysis is a promising technique for several peripheral nerves and specifically those that can aid in the treatment of significant pelvic pain, more work is needed in this area in the form of further data, ideally through multicenter trials. Specifically, data that would allow clinicians to better counsel patients on the arc of their expected pain relief, likelihood of adverse events, and durability of treatment would be beneficial. Another area in open debate is whether or not a block is warranted. Data delineating the number of patients who do not have a response and, therefore, avoid cryoneurolysis after careful selection would be of interest. Similarly, whether the nature of the pain generator, for instance, cancer versus non-cancer pain, predicts response remains an open question.

Finally, and perhaps of most pressing need, is clarification on the ablation protocol. Currently, it is unknown how long a freeze is needed for each peripheral nerve target, whether two freeze/thaw cycles are needed, and if temperature tracking provides clinical benefit. The lack of data around the freezing protocol speaks directly to the general lack of dose–response data. Other important considerations along these lines are thermal dose monitoring and probe selection. At present, providers utilize commercially available probes that are not necessarily calibrated for this purpose. Further data that evaluates this would help provide physicians and patients with guidance.

The above highlighted areas emphasize the need for prospective trials, standardized treatment protocols and more patient data to help move cryoneurolysis forward.

## 5. Conclusions

Cryoneurolysis is a promising technique for patients with refractory pain, which allows treatments to be personalized to the patient. While its complication and side effect profiles appear to be well within acceptable limits, further data is needed in this area.

## Figures and Tables

**Figure 1 jpm-15-00587-f001:**
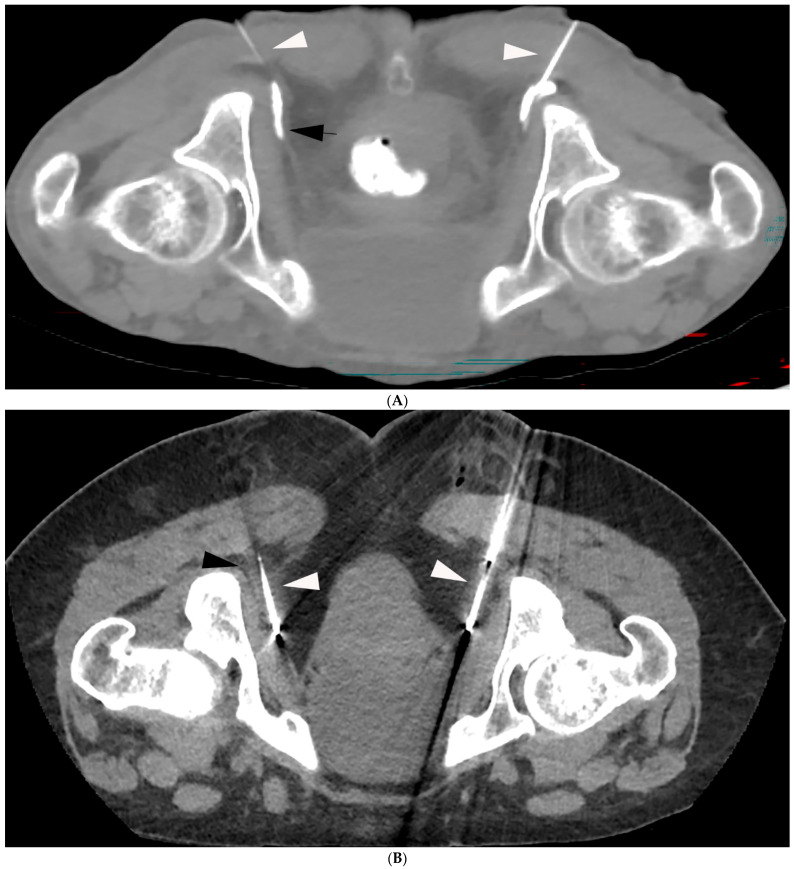
(**A**) Single non-contrast enhanced computed tomography (CT) image demonstrates two 21-gauge needles (white arrows) with their tips in the pudendal canal. Injected dilute contrast is seen to be traveling within the canal (black arrow). (**B**) A single non-contrast-enhanced CT image demonstrates cryoablation probes (white arrows) within the pudendal canals (black arrow).

**Figure 2 jpm-15-00587-f002:**
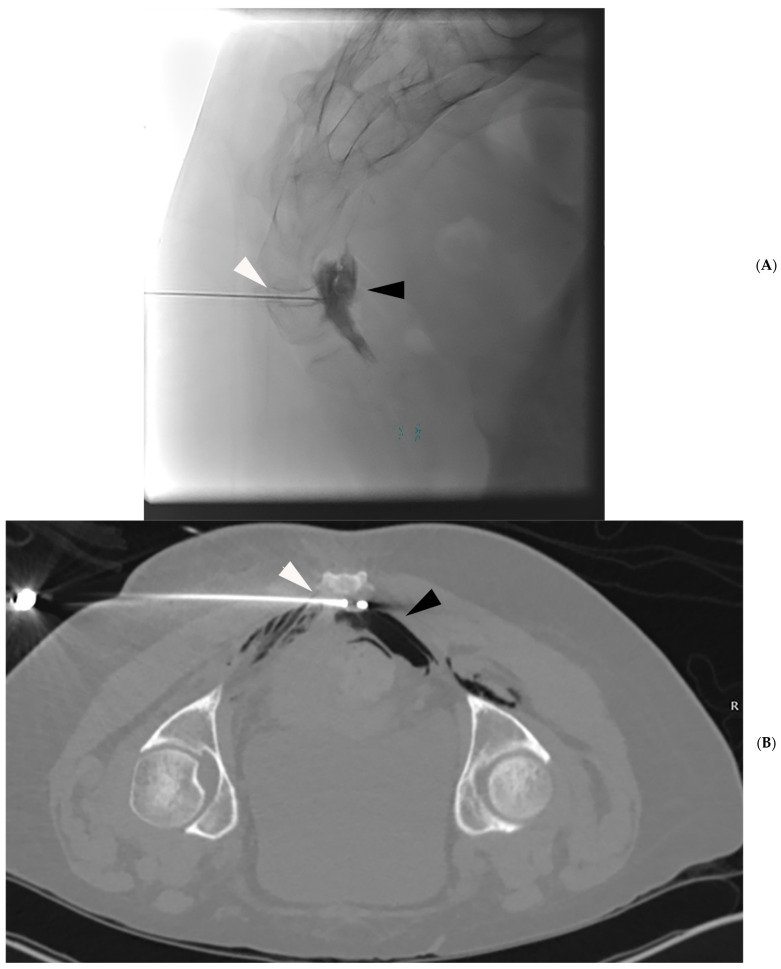
(**A**) Single fluoroscopic image demonstrates a 21-gauge needle that has been advanced through the sacrococcygeal junction (white arrow), and contrast is pooling in the region of the impar ganglion (black arrow). (**B**) A single cryoablation probe has been placed from a lateral approach just anterior to the sacrococcygeal junction (white arrow). Pneumo-dissection (black arrow) has been utilized to move the bowel away.

**Figure 3 jpm-15-00587-f003:**
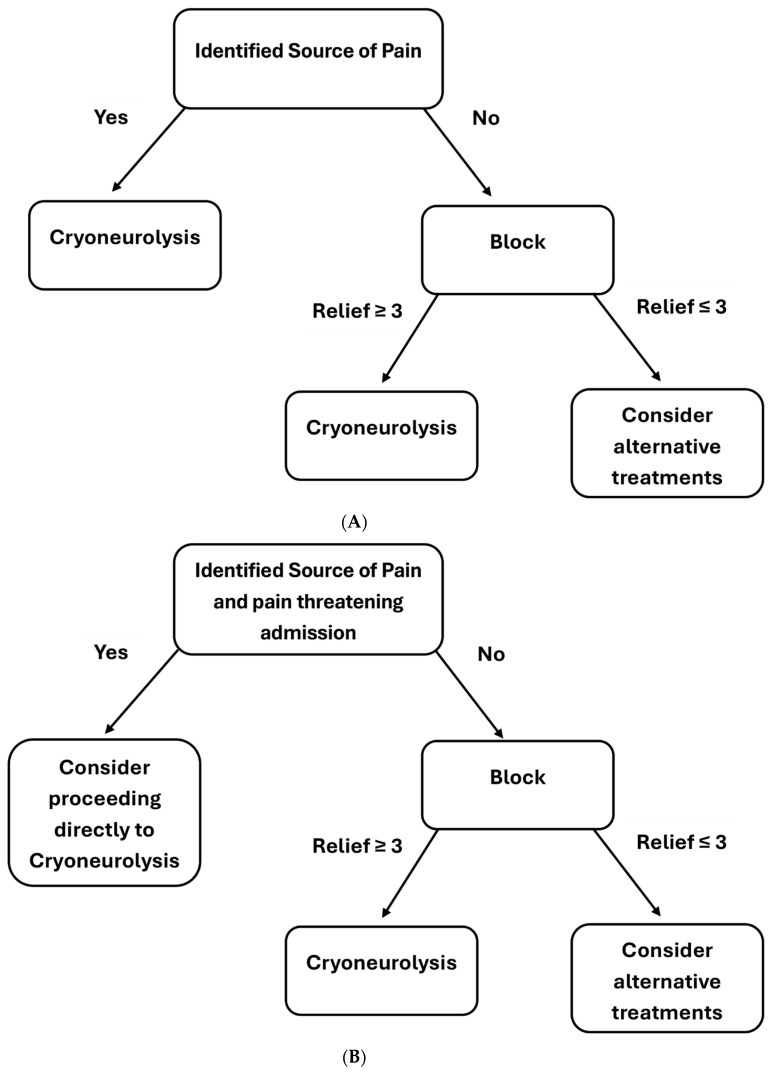
(**A**) Flow diagram demonstrating the author’s institutional protocol for treatment of inpatients with significant pelvic pain. (**B**) Flow diagram demonstrating the author’s institutional protocol for treatment of outpatient with significant pelvic pain.

**Table 1 jpm-15-00587-t001:** Summary of the available literature on pudendal cryoneurolysis.

Study	Number of Patients	Type of Pain	VAS	Adverse Events
Prologo et al. [37]	10	Cancer	5.2 ^#^	None
Prologo et al. [38]	11	Non-Cancer	Pre: 7.6 24 H post: 2.6 45 D post: 3.5 6 M post: 3.1	None
Hampton et al. [59]	2	Non-Cancer	25–85% reduction	None

VAS = visual analog scale, H = hours, D = days, M = months, ^#^ = reported mean difference between pre and post cryoneurolysis pain.

## Data Availability

Not applicable.

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
