# Peer review of "Cryoneurolysis: An Emerging Personalized Treatment Strategy for Significant Pelvic Pain"

_jpm, 2025, doi:10.3390/jpm15120587_

Round 1
Reviewer 1 Report
Comments and Suggestions for Authors
While this review introduces cryoneurolysis as a novel approach for pelvic pain management, it falls short in several key areas that undermine its scientific rigor and persuasiveness.
1) The manuscript lacks transparency in literature sourcing and selection. There is no clear search strategy, database specification, or inclusion criteria, making the review appear anecdotal rather than systematic. The authors summarize findings without critical appraisal of study quality, sample size, or bias. A structured comparison of evidence levels would lend much-needed credibility.
2) Although the mechanism of cryoneurolysis is described historically, quantitative and biophysical details are insufficient. The paper omits discussion of dose–response parameters, probe calibration, or thermal dose modeling—critical elements for reproducibility and mechanistic validity. Assertions regarding safety and procedural control are largely narrative, with no comparative outcome data.
3) The discussion presents cryoneurolysis as promising but does not adequately acknowledge the paucity of clinical evidence. The cited studies are few, small, and uncontrolled, yet the tone implies therapeutic equivalence with established modalities. Potential complications and failure mechanisms deserve a more balanced exploration.
4) Stronger emphasis on randomized controlled trials, standardized treatment protocols, and multicenter registries is essential to move beyond preliminary, anecdotal observations.
Author Response
Reviewer 1:
1) The manuscript lacks transparency in literature sourcing and selection. There is no clear search strategy, database specification, or inclusion criteria, making the review appear anecdotal rather than systematic. The authors summarize findings without critical appraisal of study quality, sample size, or bias. A structured comparison of evidence levels would lend much-needed credibility.
Reply: Thanks for the comment, this is not intended to be a meta-analysis or systematic review rather a narrative review. The authors believe that a systematic review of this emerging technique would be premature at this time.
2) Although the mechanism of cryoneurolysis is described historically, quantitative and biophysical details are insufficient. The paper omits discussion of dose–response parameters, probe calibration, or thermal dose modeling—critical elements for reproducibility and mechanistic validity. Assertions regarding safety and procedural control are largely narrative, with no comparative outcome data.
Reply: Thanks for the comment, the authors agree that the lack of data on dose-response parameters is a weakness of this emerging technology. Probe calibration is done by the manufacturer and the authors have added some discussion of this in the future directions paragraph. Similarly, some discussion of thermal dose modeling has been added to the future directions paragraph. The authors agree this is a narrative review and thus much of the discussion is narrative in nature.
3) The discussion presents cryoneurolysis as promising but does not adequately acknowledge the paucity of clinical evidence. The cited studies are few, small, and uncontrolled, yet the tone implies therapeutic equivalence with established modalities. Potential complications and failure mechanisms deserve a more balanced exploration.
Reply: The manuscript has been reworked with the reviewers comments in mind.
4) Stronger emphasis on randomized controlled trials, standardized treatment protocols, and multicenter registries is essential to move beyond preliminary, anecdotal observations.
Reply: Thanks for the comment, the authors have attempted to emphasize the reviewers valid point in the revised manuscript.
Reviewer 2 Report
Comments and Suggestions for Authors
Please read the comments presented in the review.

Author Response
- The literature review is based almost exclusively on small retrospective studies and case series (n<15), which limits the possibility of drawing far-reaching conclusions.
Reply: Thanks for the comment, the authors agree with these limitations and have reworked the manuscript with the reviewers comments in mind.
- Methodology – the literature search method (databases, keywords, publication period) has not been specified, which makes it difficult to assess the completeness of the work. The missing section should be added to the article before publication.
Reply: Thanks for the comment, this is not intended to be a meta-analysis or systematic review rather a narrative review. The authors believe that a systematic review of this emerging technique would be premature at this time.
- Research results – please summarise the research results in a table – a summary of the data contained in the publications (number of patients, type of pain, VAS before/after surgery, complications). This will increase the transparency of the work.
Reply: Thanks for the comment, table 1 has been added to the revised manuscript.
- Conclusions – please formulate your conclusions precisely and concisely – recommendations for future research (e.g. randomised controlled trials, standardisation of cryotherapy protocols).
Reply: The manuscript has been reworked with the reviewers comments in mind.
- ï‚·The quality of Figure 3 should be improved.
Reply: Figure 3 has been reworked with the reviewers comments in mind.
Round 2
Reviewer 1 Report
Comments and Suggestions for Authors
In this revised version the manuscript has much improved